# Attenuation of Hypertrophy in Human MSCs via Treatment with a Retinoic Acid Receptor Inverse Agonist

**DOI:** 10.3390/ijms21041444

**Published:** 2020-02-20

**Authors:** Moritz Riedl, Christina Witzmann, Matthias Koch, Siegmund Lang, Maximilian Kerschbaum, Florian Baumann, Werner Krutsch, Denitsa Docheva, Volker Alt, Christian Pfeifer

**Affiliations:** 1Department of Trauma Surgery, Regensburg University Medical Center, 93053 Regensburg, Germany; moritzriedl@icloud.com (M.R.); matthias.koch@ukr.de (M.K.); Siegmund.Lang@ukr.de (S.L.); maximilian.kerschbaum@ukr.de (M.K.); Florian.Baumann@ukr.de (F.B.); Werner.Krutsch@ukr.de (W.K.); volker.alt@ukr.de (V.A.); 2Laboratory of Experimental Trauma Surgery, Department of Trauma Surgery, Regensburg University Medical Center, 93053 Regensburg, Germany; christina.witzmann@stud.uni-regensburg.de (C.W.); denitsa.docheva@ukr.de (D.D.)

**Keywords:** mesenchymal stem cells, chondrogenesis, hypertrophy, BMS, retinoic acid receptor, inverse agonist

## Abstract

In vitro chondrogenically differentiated mesenchymal stem cells (MSCs) have a tendency to undergo hypertrophy, mirroring the fate of transient “chondrocytes” in the growth plate. As hypertrophy would result in ossification, this fact limits their use in cartilage tissue engineering applications. During limb development, retinoic acid receptor (RAR) signaling exerts an important influence on cell fate of mesenchymal progenitors. While retinoids foster hypertrophy, suppression of RAR signaling seems to be required for chondrogenic differentiation. Therefore, we hypothesized that treatment of chondrogenically differentiating hMSCs with the RAR inverse agonist, BMS204,493 (further named BMS), would attenuate hypertrophy. We induced hypertrophy in chondrogenic precultured MSC pellets by the addition of bone morphogenetic protein 4. Direct activation of the RAR pathway by application of the physiological RAR agonist retinoic acid (RA) further enhanced the hypertrophic phenotype. However, BMS treatment reduced hypertrophic conversion in hMSCs, shown by decreased cell size, number of hypertrophic cells, and collagen type X deposition in histological analyses. BMS effects were dependent on the time point of application and strongest after early treatment during chondrogenic precultivation. The possibility of modifing hypertrophic cartilage via attenuation of RAR signaling by BMS could be helpful in producing stable engineered tissue for cartilage regeneration.

## 1. Introduction

Mesenchymal stem cells (MSCs) are a promising source for the regeneration of articular cartilage defects. They can be easily obtained from different kinds of tissue (e.g., bone marrow [1] and adipose tissue [2]) and have a high chondrogenic potential [3,4,5,6]. Furthermore, MSCs could be able to recapitulate the embryonic lineage transitions originally involved in the formation of pheno- and genotypically stable joint tissue.

Unfortunately, chondrogenically differentiating MSCs tend to undergo hypertrophy in vitro similar to transient growth plate chondrocytes in long bones. Despite more than 20 years of research, this remains an issue that still makes them inappropriate for clinical use in articular cartilage repair [4,5,6,7,8]. In vivo chondrogenically precultured MSCs showed induction of hypertrophy, vascular invasion, and terminal matrix calcification after ectopic transplantation into mice, whereas implanted human articular chondrocytes kept a stable chondrogenic phenotype without signs of hypertrophy [9]. This indicates discrepancies in the developmental program of chondrogenically differentiated MSCs and articular chondrocytes that have to be overcome to produce appropriate repair tissue.

### 1.1. Cell Fate of Limb Chondrocytes

During embryogenesis, articular chondrocytes, as well as growth plate chondrocytes develop from common mesenchymal precursor cells descending from the lateral plate mesoderm. As an initial step of skeletogenesis, these multipotent mesenchymal stem cells condense forming the cartilaginous anlagen of the bones and differentiate into highly proliferative pre-chondrocytes [10,11]. A special subgroup of densely packed mesenchymal cells at the future joint site now builds the avascular interzone which represents the first sign of joint formation. Adjacent progenitor cells in the interrupted cartilaginous skeleton differentiate into chondrocytes that become organized to form the growth plates, undergo hypertrophy, vascular invasion, and finally ossification. Meanwhile the interzone cells give rise to the permanent articular chondrocytes for both interlocking joint surfaces [10,12,13]. Each cell population is characterized by distinct expression patterns [14,15].

### 1.2. Retinoic Acid Receptor Signaling

Several studies attribute the retinoic acid receptor (RAR) pathway as having an important influence on the cell fate of mesenchymal progenitors following a distinct spatiotemporal pattern [16,17,18,19]. Suppression of RAR signaling by unliganded RAR seems to be a crucial requirement for chondrogenesis and expression of prochondrogenic genes and growth factors including Sox genes and BMPs [16,17,18,19]. Accordingly, retinoids have been shown to attenuate chondrogenesis in the early stage of differentiation [20,21,22]. This fact contrasts the avascular constitution of articular cartilage with the vascular invasion during endochondral ossification of the growth plate, as endogenous retinoids are mainly distributed through the blood vessel system. The subsequent ligand-dependent activation of RAR signaling in the late stage of growth plate development induces hypertrophy and mineralization [23,24,25,26]. RAR signaling is transmitted by the following two subfamilies of intranuclear retinoid receptors: The retinoic acid receptors, RARα, -β, and -γ, and the retinoic X receptors, RXRα, -β, and -γ [27,28]. RAR and RXR form heterodimers and modulate target gene expression operating as ligand-dependent transcription factors. The RARs interact with different coregulators and depending on the binding ligand, bonds to coactivators (CoA) or corepressors (CoR) are reinforced or weakened [29]. Non-liganded receptor complexes exhibit basal repression and are involved in chromatin condensation, which inhibits gene transcription [30,31]. Activation of RAR signaling by the physiological agonist all-trans retinoic acid (tRA) induces target gene expression by recruitment of CoA and, then, activates further downstream pathways including the *Wingless/Int* (Wnt)/β-catenin pathway [29,32]. Wnt/β-catenin signaling then initiates hypertrophic conversion through upregulation of runt-related transcription factor 2 (Runx2) [33] and, consequently, increased MMPs and collagen type X expression [34]. In contrast, Sox9 [35], and therefore expression of chondrogenic genes, such as aggrecan and collagen type II, was decreased [36]. Figure 1 describes a more detailed illustration of the RAR pathway and its downstream signaling.

As hypertrophy would result in apoptosis and ossification, a stable and functional tissue engineered MSC based cartilage implant for cartilage defects remains to be elucidated.

Wael et al. showed that chondrogenesis in MSCs can be induced by treatment with a neutral RAR antagonist that showed less signs of hypertrophy as compared with common chondrogenesis protocols [42].

We hypothesized that repression of RAR signaling is capable of attenuating hypertrophy, and thus treating chondrogenically differentiating MSCs with a pan-RAR inverse agonist, BMS204,493. As an inverse agonist (IA), BMS must be distinguished from neutral antagonists that are only capable of competitive replacement of agonists. BMS causes a more active process and increases CoR interactions as compared with the unliganded receptor state [29,43,44]. Figure 2 illustrates the expected effect of BMS on RAR signaling.

Since the effect of retinoic acid in growth plate chondrogenesis depends on the state of differentiation, we further investigated if BMS application at different time points leads to different outcomes.

In our study design, tRA lacks a competitive rival for BMS at RAR as we focus on the unimpaired effect of BMS. Unliganded RAR acts prochondrogenically by basal repression of target gene transcription [31] and inverse agonists such as BMS are capable of further reducing basal receptor activity in the absence of the physiological agonist [43].

Additionally, we investigated the effect of the RAR agonist tRA on hypertrophic conversion to gain a better understanding of RAR signaling in chondrogenically differentiating hMSCs.

## 2. Results

### 2.1. Induction of Hypertrophy

After 28 days of cell culture under chondrogenic conditions, aggregates demonstrated a homogenous morphology consisting of small round cells that were surrounded by a uniform ECM very similar to hyaline cartilage structure (Figure 3). In comparison, under hypertrophy enhancing conditions, MSCs exhibited a very pronounced hypertrophic phenotype with high cell volume and large intracellular lacunae specifically in the outer regions of the aggregates. The number and size of hypertrophic cells, but not the number of cells overall within the aggregates, are statistically significantly increased under hypertrophic medium conditions as compared with chondrogenic control aggregates (Figure 4). Although GAGs and collagen type II are typical chondrogenic markers, DMMB staining for GAGs and immunohistochemical staining for collagen type II are strong in hypertrophic MSC aggregates as well. Analogous to that, GAG content (Figure 5B) and collagen type II gene expression (Figure 6B) are high in hypertrophic groups as compared with the chondrogenic groups.

Collagen type X staining (Figure 3) illustrates distinct differences between chondrogenic and hypertrophic groups. While chondrogenic aggregates feature only slight staining, collagen type X, a primary constituent in hypertrophic chondrocyte ECM [4], is clearly increased under hypertrophic conditions and spread over almost the whole aggregate. Consistently, collagen type X gene expression (Figure 6D) was upregulated under hypertrophic conditions as compared with the chondrogenic group.

As alkaline phosphatase (ALP), which initiates enzymatic ECM mineralization, is a reliable biochemical marker for hypertrophy and in later stages for osteoblast activity, ALP staining (Figure 3) is highly expressed in hypertrophic areas and at the edge of the hypertrophic aggregates but limited to a thin frame around the aggregates under chondrogenic conditions. These findings are in accordance with the ALP activity analysis in medium supernatant on Day 28 of MSC pellet culture (Figure 5A). ALP activity is significantly higher in hypertrophic control aggregates as compared with chondrogenic control aggregates.

MMP13, a metalloproteinase that degenerates ECM in chondrocytes before ossification, reveals higher gene expression levels under hypertrophy enhancing conditions as compared with chondrogenic MSC pellets (Figure 6C). Furthermore, gene expression of collagen type I (Figure 6A) is negligible at every time point and medium condition, which actually is in agreement with previous studies [45,46,47], thus, demonstrating that collagen type I expression is high in undifferentiated mesenchymal cells and rapidly decreases in a very early stage of chondrogenic differentiation.

### 2.2. Enhancement of Hypertrophy by RA Treatment

Treatment with RA increases the hypertrophic phenotype in MSC aggregates (Figure 3). Treated cells are even bigger than that in hypertrophic control aggregates and cellular lacunae dominate the histological appearance of the entire aggregates. The time point of application during cell culture does not seem to influence the effect of RA on chondrogenically differentiating hMSCs.

DMMB staining is weaker under RA influence as compared with the hypertrophic control, and BMS treated aggregates indicate a lower content of GAGs. Biochemical analyses confirm a clearly reduced GAG/DNA ratio under RA treatment as compared with the chondrogenic control, as well as BMS treated groups (Figure 5B). Similar to the other hypertrophic aggregates, immunohistochemical staining (Figure 3) shows a high content of collagen type II in RA groups, both in the early and the late treatment group. However, collagen type II gene expression (Figure 6B) is downregulated as compared with the hypertrophic control and especially with the BMS treated groups. Collagen type X staining (Figure 3) appears homogenous in the whole aggregate under RA treatment but is slighter as compared with the hypertrophic control group and the late BMS group. Under RA treatment, collagen type X gene expression (Figure 6D) is downregulated as compared with the hypertrophic control and even lower than in the early BMS group. In RA treated groups, ALP staining (Figure 3) shows strong enzyme activity at the edge of the aggregates but slight staining in the center. The measurement of ALP activity (Figure 5A) in medium supernatant of RA treated aggregates shows a distinct increase in enzyme activity as compared with the chondrogenic control but no meaningful differences as compared with the hypertrophic groups either the control or the BMS treated groups. Gene expression of MMP13 (Figure 6C) is upregulated by RA treatment under hypertrophy enhancing conditions.

### 2.3. Attenuation of Hypertrophy by BMS Treatment

Attenuation of hypertrophy by BMS depends on the time point of application during pellet culture. Application of BMS in the late phase of chondrogenesis does not result in clear morphological changes (Figure 3). However, there are distinct differences in the groups undergoing early BMS treatment. Apart from single hypertrophic cells, the aggregates have a very homogenous structure consisting of small chondrocytes and an ECM enriched in GAGs. The number of hypertrophic cells is statistically significantly decreased by early BMS treatment under hypertrophic conditions and the hypertrophic chondrocytes are significantly smaller under both early, as well as late, BMS treatment (Figure 4). Histologically, chondrogenic markers appear not to be influenced by BMS treatment. DMMB staining and immunohistochemical staining for collagen type II are equally strong in BMS groups as compared with the hypertrophic control group. Although collagen type II synthesis on protein level stays unaffected by BMS treatment, COL2A1 gene expression (Figure 6B) under hypertrophic medium conditions is increased. In doing so early, BMS treatment reveals the highest expression levels of COL2A1. Under hypertrophy enhancing conditions, BMS treatment increases GAG content (Figure 5B). Biochemical determination of GAG content in the ECM confirms histological results and shows similar levels in BMS treated aggregates as compared with hypertrophic control aggregates.

The synopsis of our results presents a reduction of hypertrophic markers under BMS treatment. Collagen type X staining (Figure 3) is limited to the edge of the aggregates under late BMS treatment and is reduced even more in aggregates that have obtained BMS during the first two weeks of pellet culture. Although the distinct histological results cannot be supported by statistically significant differences of collagen type X expression (Figure 6D) between the hypertrophic groups, gene expression of collagen type X in hypertrophic groups tends to be reduced under BMS treatment. ALP staining (Figure 3), after late BMS treatment, is reduced in the center and the outer zone of aggregates as compared with the hypertrophic control but still strong in the intermediate layer. Early BMS application decreases ALP staining even more. Analogously, there is also a slight reduction of ALP activity (Figure 5A) in the hypertrophic groups treated with BMS, both early and late BMS treatment group. Furthermore, gene expression of MMP13 (Figure 6C) is downregulated upon treatment with BMS as compared with the hypertrophic controls.

## 3. Discussion

In our study, we initially confirmed the findings of previous works [8,48,49] that special culture conditions, including addition of BMP4, induce hypertrophy in chondrogenic differentiating hMSCs on histological, biochemical, and genetical levels. Activation of the RAR pathway by the addition of tRA further increases the hypertrophic conversion with upregulated MMP13 gene expression and reduced GAG content and COL2A1 expression. Only collagen type X gene expression deviates from the sample, as it is lowest in RA treated groups among all hypertrophic groups. We think there is a connection in these incongruent results with the complex timetable of RAR signaling during chondrogenic differentiation, i.e., with respect to the time point of RA application during cell culture, as the effect of BMS treatment is time-course dependent. Thus, for the future, one of our major aims would be to further investigate this issue to achieve an optimum outcome.

Inhibition of RAR signaling in chondrogenic differentiating hMSCs by treatment with the pan-RAR inverse agonist BMS clearly attenuates hypertrophic conversion, as shown by a decreased number and size of hypertrophic cells, reduced collagen type X staining and lower ALP presence. The efficacy of BMS treatment under hypertrophic conditions is dependent on the time point of application in the pellet culture. The BMS treatment in the late phase of chondrogenesis only shows little reduction in the volume and number of hypertrophic cells, whereas the group with the early BMS treatment shows distinct morphological differences. Apart from single hypertrophic cells, the aggregates have a very homogenous structure consisting of small chondrocytes. In comparison, activation of RAR signaling via RA treatment produces an opposite effect and leads to even larger hypertrophic chondrocytes. Regarding the morphology of MSCs, this directly mirrors the expected role of RAR signaling in hypertrophy.

Staining of hypertrophic markers, ALP and collagen type X, is clearly reduced in both the late BMS group and even more in the early BMS group. In summary, aggregates that have obtained BMS during the first two weeks of the pellet culture show the best histological results regarding the anti-hypertrophic effect. On the opposite side, direct activation of the RAR pathway by application of retinoic acid leads to enhancement of the hypertrophic phenotype.

Collagen type II staining and gene expression is at an unexpected high level in hypertrophic groups, as it is actually a chondrogenic marker, which physiologically decreases in reverse order to collagen type X content. This phenomenon is founded on the effect of BMP4 in the proliferation medium to enhance hypertrophy. BMP4 is supposed to not only stimulate chondrocyte maturation and osteogenesis, but also to induce initial steps of chondrogenesis and guide mesenchymal cells towards expression of cartilage differentiation markers [50,51,52]. Thus, we assume that BMP4 in our model initially boosts the entire synthesis activity of the chondrogenically differentiating MSCs and especially the production of chondrogenic markers. Collagen type II concentration would probably decrease during a longer duration, as MMPs and collagenase degrade collagen type II over time. However, inhibition of RAR signaling by BMS even further increased collagen type II gene expression in hypertrophic aggregates, which could also be a sign for the prochondrogenic effect of BMS on MSCs and a synergism of BMP4 and BMS in chondrogenesis. These results confirm the findings of Hoffman et al. who showed that RA and BMPs act as opponents in early chondrogenesis and silencing of RAR signaling is required for the chondrogenic stimulatory activity of BMP4 [16]. Furthermore, several studies have shown that RAR and BMP signaling are closely associated but also indicate that they cooperate on various levels. Furthermore, apart from the direct retinoid effects on chondrocyte maturation through activation of RARs, retinoic acid seems to stimulate genes encoding BMPs which leads to an indirect induction of collagen type X expression in pre-hypertrophic chondrocytes [23,24]. According to these findings, RAR signaling in the late stage of chondrocyte maturation seems to be at least partially mediated by BMPs. In return, BMP4 treatment increases the gene expression of specific RAR and RXR subtypes [53]. With respect to these interactions between the RAR and the BMP pathway, it appears reasonable that inhibition of RAR signaling with BMS has an influence on BMP4 effects. Whether the effect is synergistic or antagonistic could be dependent on the kind of cells and the state of differentiation.

Gene expression of hypertrophic markers MMP13 and collagen type X is reduced by BMS treatment especially upon application at the earliest stages of differentiation as evidence for the prochondrogenic effect of BMS. In regard to MMP13 gene expression, the effect of RAR signaling becomes even clearer by having a look at the RA treated groups. RA, in contrast to BMS, leads to a strong upregulation of MMP13 gene expression as compared with the hypertrophic control.

As a sign for increased synthesis activity, the high GAG content in hypertrophic group, similar to the high collagen type II gene expression, can probably be traced back to the effect of BMP4. The results in BMS treated groups turned out to be inconsistent, as GAG content is increased in the late treated group, illustrating an enhanced chondrogenic differentiation, whereas early BMS treatment leads to a lower GAG content. This result is surprising in view of a former analysis, as early BMS treatment was found to be more effective than late treatment inhibiting hypertrophy and improving the chondrogenic phenotype. However, as a more representative result that demonstrates the effect of RAR signaling on chondrogenesis, application of RA leads to a distinct decrease in GAG content as compared with BMS treated aggregates, which is also reflected by a reduction in DMMB staining.

We found a statistically significant higher ALP activity in the hypertrophic control group as compared with the chondrogenic control group, but the BMS treatment in both early and late phase of differentiation reduces ALP activity under hypertrophy enhancing conditions down to the range of the chondrogenic groups. Although we saw a distinct trend towards a prochondrogenic effect of BMS treatment on differentiating MSCs, gene expression and ALP activity results did not reach statistical significance. This could be due to the number of donors but also to the generally high heterogeneity of MSCs among different donors and cell populations [54] and to the differences due to donor age and gender [55].

Despite promising in vitro results, there are limitations for the in vivo use of BMS. Indeed, retinoids are clinically used for a number of therapeutic indications including cancer, psoriasis, acne, and diabetes but they can lead to severe side effects especially through systemic application. Thus, the topical application by intraarticular injection would be favorable. For the therapy of osteoarthritis, Yin et al. developed a drug delivery system especially for BMS using an engineered cartilage oligomeric matrix protein coiled-coil protein to encapsulate and protect the hydrophobic and unstable BMS molecule [56].

Furthermore, there are novel atypical drugs that could be an alternative to BMS in vivo. Busby et al. presented the first nonacid, non-retinoid direct modulator of the RAR superfamily that acted as a pan-RAR inverse agonist, similar to BMS, but featured an improved toxicity and pharmacokinetic profile over classical retinoids [57]. However, their efficiency in chondrogenesis protocols has to be further investigated.

An important aim for the future is to further analyze the exact processes and sequences involved in RAR signaling during MSC chondrogenesis. Current studies are already investigating target genes of the RAR complex and downstream pathways, in particular the Wnt/β-catenin signaling, as illustrated in Figure 1.

## 4. Material and Methods

### 4.1. Isolation of MSCs

Human MSCs were isolated from bone marrow aspirates of the iliac crest of patients undergoing surgery that required autologous bone grafting with approval of the ethics committee of the local university and informed written consent. Five patients between 23 and 32 years and one patient at the age of 59 were included in the study (*n* = 6). MSCs were isolated by Ficoll^®^ (Biochrom, Berlin, Germany) gradient centrifugation. Cells were expanded in Dulbecco’s modified Eagle’s medium (DMEM) low glucose (Invitrogen, Carlsbad, CA, USA) with 10% fetal calf serum (PAN Biotech GmbH, Aidenbach, Germany) and 1% penicillin/streptomycin (Invitrogen) at 37 °C with 5% CO_2_ The medium was changed twice a week and cells were trypsinized at 80% confluence and frozen for later use in liquid nitrogen. After thawing and monolayer expansion, cells were used for the experiments at Passage 2.

### 4.2. Chondrogenic Differentiation and Hypertrophic Conversion upon Treatment with BMS493 and RA

On the basis of the works by Karl et al. [48], who proved that hypertrophy in chondrocytes is significantly mediated by bone morphogenic protein (BMP) 4, we used an in vitro hypertrophy model for MSCs. In this model hypertrophy in chondrogenically differentiating MSCs can be increased by withdrawal of TGFβ and dexamethasone and the addition of BMP4. This hypertrophic challenge clarifies the impact of the anti-hypertrophic treatment in a distinct manner.

MSCs were trypsinized and seeded in V-bottomed 96-well polypropylene plates at 200,000 cells per well. Pellets were formed by centrifugation at 250 g for 5 min and chondrogenically predifferentiated for 14 days in the DMEM with high glucose (Invitrogen), 1% ITS (Sigma-Aldrich), 50 mg/mL ascorbate-2-phosphate (Sigma-Aldrich, St. Louis, MO, USA), 40 mg/mL l-proline (Sigma-Aldrich), 100 nM dexamethasone (Sigma-Aldrich), 1 mM sodium pyruvat (Invitrogen), and 10 ng/mL TGFβ1 (R&D Systems). After a predifferentiation period of 14 days, medium conditions were changed and from this time point, one part of the groups maintained an hypertrophy-enhanced medium consisting of DMEM high glucose, 1% ITS, 50 g/ mL ascorbate-2-phosphate, 40 g/mL l-proline, and 25 ng/mL BMP4 (R&D Systems). Additionally, certain hypertrophic groups were treated with BMS204,493, (2 μM, Sigma), others with RA (10 nM, Sigma) at different time points (see Table 1). The medium was changed three times per week. Aggregates were harvested on Day 28 for gene expression analysis and biochemical and histological examination. The medium supernatant was collected for determination of ALP activity.

### 4.3. Histological and Immunohistochemical Analysis

Frozen sections of aggregates were stained with dimethylmethylene blue (DMMB) (Sigma-Aldrich). Histochemical ALP staining was performed with an alkaline phosphatase kit (Sigma-Aldrich) with neutral red as counterstaining. For immunohistochemistry, the following antibodies were used: Mouse anti-collagen type X (1:20; Quartett Immunodiagnostika und Biotechnologie GmbH, Berlin, Germany) and mouse anticollagen type II (1:100; Calbiochem (Merck), Darmstadt, Germany). After blocking of endogenous peptidases (3% H_2_O_2_/10% methanol in PBS) for 30 min, sections were incubated in a blocking buffer (10% fetal bovine serum/10% goat serum in PBS) for 60 min at room temperature (RT) followed by incubation with an appropriate primary antibody in the blocking buffer overnight at 4 °C. For collagen type II and type X staining, antigen retrieval with pepsin digestion for 15 min at RT and for collagen type X staining, additional hyaluronidase digestion for 60 min at RT were performed before blocking. Immunolabeling was detected with a biotinylated secondary antibody (1:100; Dianova, Hamburg, Germany), horseradish peroxidase-conjugated streptavidin (Vector Laboratories), and metal enhanced diaminobenzidine as substrate (Sigma-Aldrich). For each antibody, a negative control without a primary antibody was conducted. To investigate the stained sections, we used the Nikon ECLIPSE TE 2000-U microscope. On the basis of the DMMB stained sections, we determined the number of hypertrophic cells within an aggregate using ImageJ software. Sections of three aggregates per donor were analyzed. Binary images of the sections were generated and inverted so that hypertrophic cells appeared as black particles on the white background. The number of cells and the area of the aggregate were identified and put in relation to each other. The cell size of hypertrophic chondrocytes was determined in collagen type II stained sections. Therefore, images in 40 times magnification were recorded and 3 representative areas per aggregate and donor were analyzed using ImageJ.

### 4.4. RNA Isolation, cDNA Synthesis, and Gene Expression Analysis

For each donor, eight to ten aggregates per condition were pooled, homogenized using the Precellys24 homogenizer (bertin instruments, Montigny-le-Bretonneux, France), and RNA was isolated using the RNeasy Plus Universal Kit (Qiagen, Hilden, Gemany). RNA was reverse transcribed in cDNA with the Transcriptor First Strand cDNA Synthesis kit (Roche, Basel, Switzerland) according to the manufacturer’s instructions. Quantitative real-time polymerase chain reaction (PCR) was performed with Brilliant SYBR Green QPCR mix (Stratagene, San Diego, CA, USA) and a real-time PCR detection system (CFX96, Biorad, Hercules, CA, USA). Gene expression was normalized to following reference genes using the delta-Ct method: Vacuolar protein sorting 29 (VPS29), proteasome subunit beta type 4 (PSMB4), and receptor expression enhancing protein 5 (REEP5). For relativization, gene expression was also determined in cells on Day 0 directly after monolayer culture. Primer sequences are shown in Table 2.

### 4.5. ALP Activity

ALP activity was determined densitometrically by measuring the change in the absorbance at 405 nm through the transformation of p-nitrophenyl phosphate to p-nitrophenol and inorganic phosphate [58]. Therefore, the medium supernatant of the single aggregates was isolated on Day 28 and centrifuged for 5 min at maximum speed. The, 75 μL of supernatant was transferred to a 96-well flat bottom plate and 75 μL substrate solution (4 mg/mL p-nitrophenol phosphate (Sigma-Aldrich) in 1.5 M Tris, 1 mM ZnCl_2_, 1 mM MgCl_2_, pH 9.0) was added. Continuous absorbance at 405 nm was measured spectrophotometrically in a Genius plate reader (Tecan) at room temperature. The change in A_405_ over time (dA/min) was calculated in the linear range of the reaction.

### 4.6. GAG Analysis

Sulfated glycosaminoglycan (GAG) content relativized to DNA content was used as a quantitative marker for chondrogenic differentiation. For GAG analysis, 4 to 5 aggregates were harvested on Day 28 and digested in 200 μL papain digestion solution (150 μg/mL Papain (Sigma) in PBS, 6 mM L-Cystein (Merck, Darmstadt, Germany), 6 mM EDTA (Sigma), pH 6.0) at 60 °C, overnight. GAG content was determined with the DMMB method and chondroitin sulfate A from bovine trachea (Sigma-Aldrich) was used as a standard. DNA content was measured with the Quant-it Pico Green dsDNA-Kit (Invitrogen) according to the manufacturer’s instructions.

### 4.7. Statistical Analysis

Data from RT-PCR analysis, ALP activity test, and GAG content analysis are expressed as mean value ± standard deviation (SD). Each experiment was carried out using cells of three to five individual marrow preparations from different donors, as indicated in the respective experiments. The one-way ANOVA Test in SPSS statistic software (IBM) in combination with the Bonferroni post hoc test was used for statistical analysis and *p* < 0.05 was considered significant.

All methods had been previously described in part [48].

## 5. Conclusions

In our study, we demonstrated that inclusion of the RAR inverse agonist BMS493 during in vitro chondrogenesis of hMSCs can attenuate hypertrophic changes, which could be useful in producing stable engineered tissue for cartilage regeneration. Specifically, when BMS was administered under hypertrophic conditions, a phenotypical reduction in hypertrophy was observed supported by decreased cell size, number of hypertrophic cells, and collagen type X deposition. Gene expression and ALP activity analysis did not reach statistical significance but tended towards a prochondrogenic effect of BMS. Our experiments reveal the early phase of chondrogenesis as the best period for the application of BMS in order to attenuate hypertrophic conversion in chondrogenic differentiating MSCs. 

## Figures and Tables

**Figure 1 ijms-21-01444-f001:**
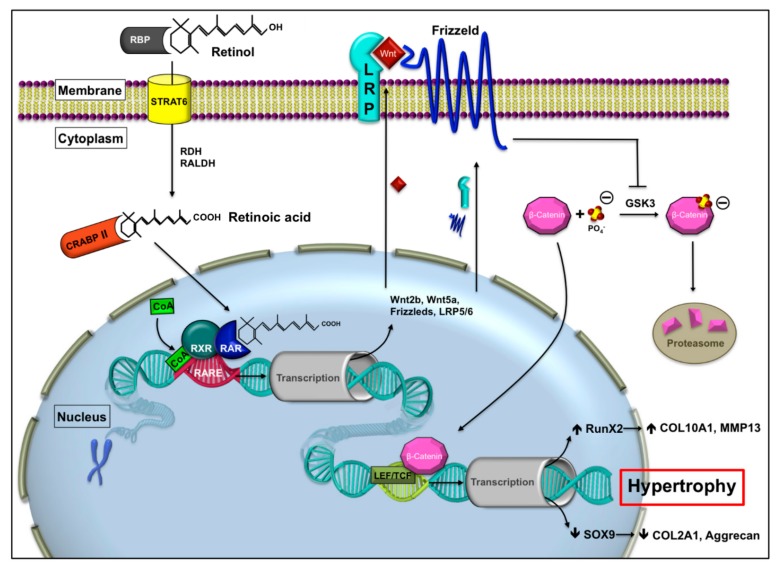
Potential interactions between retinoid signaling and the Wnt/β-catenin pathway on the cellular level for growth plate chondrocytes during endochondral ossification. The inactive retinol is transported to the cell by retinoid binding protein and translocated into the cytoplasm by the transporter STRAT6, where it is transformed into the active retinoic acid (RA) [19,37,38]. The retinoic acid is translocated into the nucleus by the cellular retinoic acid binding protein II (CRABP II) [39,40]. Binding of retinoic acid (RA) to the RA receptor would activate gene expression of Wnt proteins, receptors, and coreceptors which leads to an increased Wnt/β-catenin signaling followed by hypertrophic conversion [41]. RBP, retinoid binding protein; STRAT6, ”stimulated by retinoic acid“, receptor/Vit. A transporter; RDH, retinol dehydrogenase; RALDH, retinaldehyde dehydrogenase; RAR, retinoic acid receptor; RXR retinoic X receptor; CRABP II, cellular retinoic acid binding protein; RARE, retinoic acid response element; GSK3, glycogen synthase kinase 3 CoA co-activator; LRP, lipoprotein receptor related protein; LEF/TCF, lymphoid enhancer binding factor/transcription factor, ↑, upregulation, ↓, downregulation.

**Figure 2 ijms-21-01444-f002:**
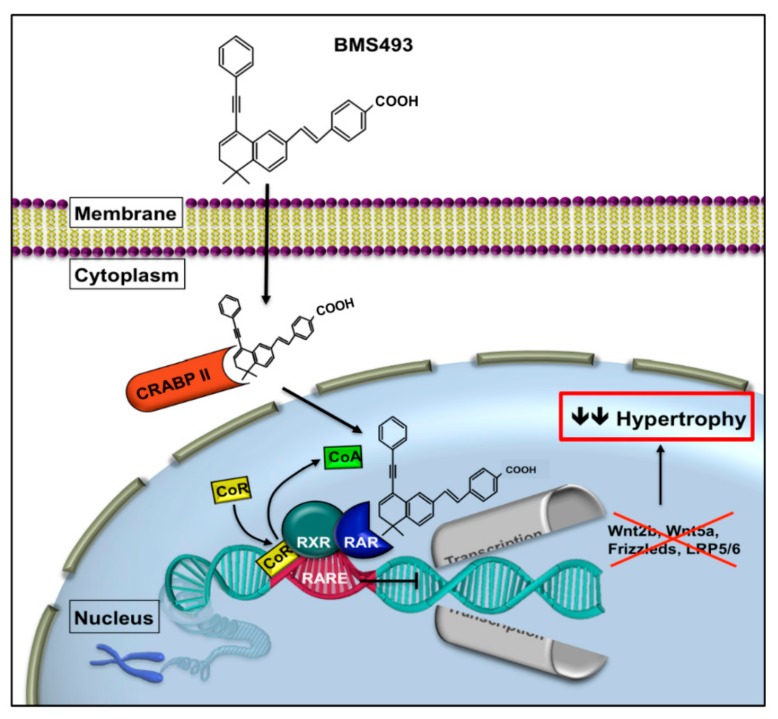
Schematic demonstration of the inhibition of the RAR pathway by BMS. BMS is translocated into the nucleus by the cellular retinoic acid binding protein II (CRABP II) and binds to the RAR/RXR complex. Binding of the inverse agonist supports corepressor recruitment. The receptor complex subsequently inhibits target gene expression at the promoter area RARE. The consequently reduced expression of Wnts and Wnt receptors and coreceptors decreases hypertrophic differentiation [29,43,44]. RAR, retinoic acid receptor; RXR, retinoic X receptor; CRABP II, cellular retinoic acid binding protein; RARE, retinoic acid response element; CoA, co-activator; CoR, co-repressor; LRP, lipoprotein receptor related protein; LEF/TCF, lymphoid enhancer binding factor/transcription factor, ↓↓ reduction.

**Figure 3 ijms-21-01444-f003:**
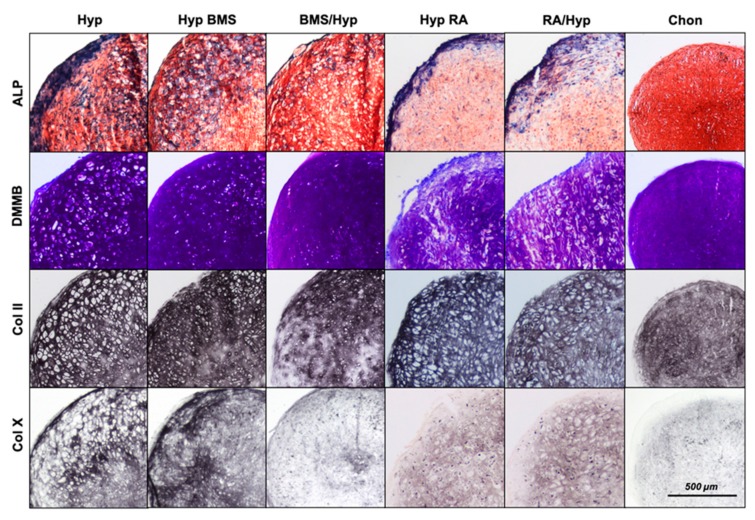
Effect of BMS and RA to the histological appearance of mesenchymal stem cells (MSC) aggregates on Day 28. First line ALP staining; second line DMMB staining; third line immunohistochemical collagen type II staining; fourth line immunohistochemical collagen type X staining. Scale bar = 500 μm. ALP, alkaline phosphatase; DMMB, dimethylmethylene blue; Coll II, collagen type II; Coll X, collagen type X; Chon, chondrogenic control group; Hyp, hypertrophic control group; Hyp BMS, hypertrophic group with late BMS treatment; BMS/Hyp, hypertrophic group with early BMS treatment; Hyp RA, hypertrophic group with late retinoic acid treatment; RA/Hyp, hypertrophic group with early retinoic acid treatment.

**Figure 4 ijms-21-01444-f004:**
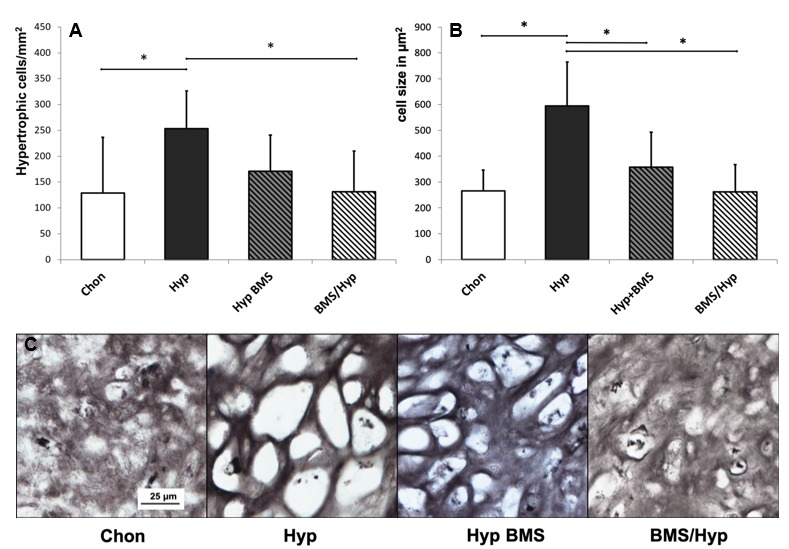
Effect of BMS on the number and size of hypertrophic cells. Image morphological determination of (**A**) number of hypertrophic cells relative to aggregate area; and (**B**) average cell size of hypertrophic cells in histological sections; (**C**) 40x magnification of the immunohistochemical collagen type II stained histological sections for illustration of hypertrophic chondrocytes under BMS treatment. *n* = 3. Whiskers represent standard deviations. Significant differences (* *p* < 0.05) are designated by asterisks. Scale bar = 25 μm. Chon, chondrogenic control group; Hyp, hypertrophic control group; Hyp BMS, hypertrophic group with late BMS treatment; BMS/Hyp, hypertrophic group with early BMS treatment.

**Figure 5 ijms-21-01444-f005:**
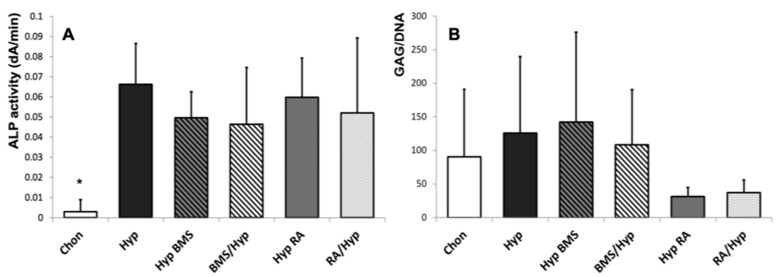
Effect of BMS and RA on ALP activity and GAG content. ALP Enzyme activity is measured densitometrically in cell culture supernatants via conversion of an ALP substrate on Day 28 (**A**); GAG content of MSC pellets on Day 28 is normalized to DNA content (**B**). *n* = 3 MSC donor populations in groups obtaining RA resp. *n* = 6 populations in other groups. Whiskers represent standard deviations. Significant differences (* *p* < 0.05) are designated by asterisks. Chon, chondrogenic control group; Hyp, hypertrophic control group; Hyp BMS, hypertrophic group with late BMS treatment; BMS/Hyp, hypertrophic group with early BMS treatment; Hyp RA, hypertrophic group with late retinoic acid treatment; RA/Hyp, hypertrophic group with early retinoic acid treatment.

**Figure 6 ijms-21-01444-f006:**
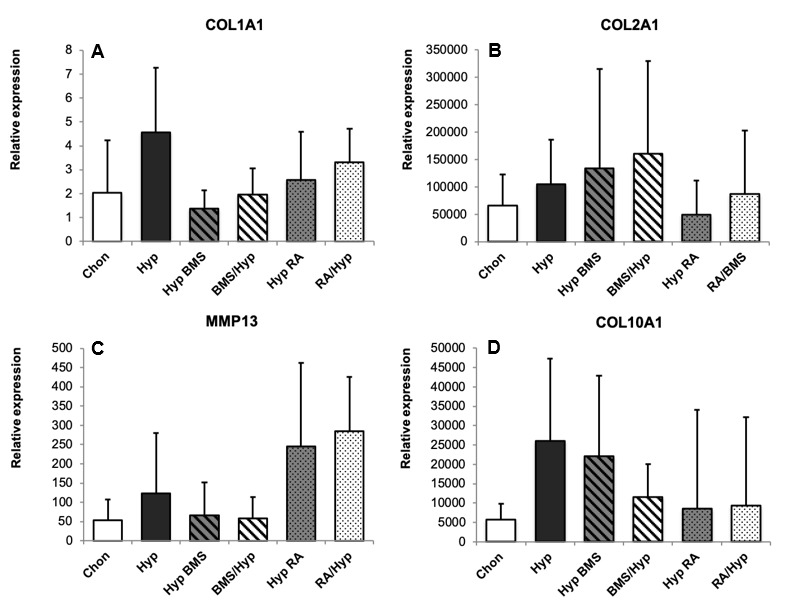
Effect of BMS and RA on gene expression. Quantitative PCR of the osteogenic marker collagen type I (**A**); the chondrogenic marker collagen type II (**B**); and the hypertrophic markers, MMP13 (**C**); and collagen type X (**D**), in MSC aggregates on Day 28. Relative gene expression normalized to Day 0 is given for *n* = 3 MSC donor populations in groups obtaining RA resp. *n* = 6 populations in other groups. Gene abbreviations are according to the NCBI database. Whiskers represent standard deviations. Chon, chondrogenic control group; Hyp, hypertrophic control group; Hyp BMS, hypertrophic group with late BMS treatment; BMS/Hyp, hypertrophic group with early BMS treatment; Hyp RA, hypertrophic group with late retinoic acid treatment; RA/Hyp, hypertrophic group with early retinoic acid treatment.

**Table 1 ijms-21-01444-t001:** Classification of aggregates into different test and associated control groups with different composition of proliferation medium.

Group	Week 1+2	d14	Week 3+4	d28
1	Chondrogenic medium	Chondrogenic medium		Chon
2	Chondrogenic medium	Hypertrophic medium		Hyp
3	Chondrogenic medium	Hypertrophic medium + BMS		Hyp w BMS
4	Chondrogenic medium + BMS	Hypertrophic medium		BMS/Hyp
5	Chondrogenic medium	Hypertrophic medium + RA		Hyp w RA
6	Chondrogenic medium + RA	Hypertrophic medium		RA/Hyp

**Table 2 ijms-21-01444-t002:** List of primers. Genes are abbreviated according to the NCBI gene database.

Gene	Sequence (Forward)	Sequence (Reverse)	Concentration
VPS29	AGCTGGCAAACTGTTGCAC	GACGGTGGTGGTGACTGAG	200 nM
PSMB4	GCTTAGCACTGGCTGCTTCT	GGACATGCTTGGTGTAGCCT	200 nM
REEP5	AGGTCAGCCACTGGGTATCA	CCTCTCTCCTCTGCAACCTG	200 nM
MMP13	GACTGGTAATGGCATCAAGGGA	CACCGGCAAAAGCCACTTTA	200 nM
COL1A1	ACGTCCTGGTGAAGTTGGTC	ACCAGGGAAGCCTCTCTCTC	200 nM
COL2A1	GGGCAATAGCAGGTTCACGTA	TGTTTCGTGCAGCCATCCT	200 nM
COL10A1	CCCTCTTGTTAGTGCCAACC	AGATTCCCAGTCCTTGGGTCA	200 nM

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
