# Peer review of "Attenuation of Hypertrophy in Human MSCs via Treatment with a Retinoic Acid Receptor Inverse Agonist"

_ijms, 2020, doi:10.3390/ijms21041444_

Round 1
Reviewer 1 Report
This manuscript addresses the issue of hypertrophic phenotype in MSCs undergoing osteogenic differentiation. The manuscript is clearly written and the authors follow a logical workflow to show their results. However, there are some points to be considered in order to improve the quality of the work. The claimed effects of RAR pathway activation/inhibitions can be appreciated in the histological analysis of samples, but the other approaches (gene expression, ALP activity, GAG/DNA) do not support those findings in a convincing manner. Authors should address and discuss the following points to improve the quality of their statements.
Comments:
1) Page 9, lane 311, MSCs donors are “under the age of 60”. This is quite imprecise, pleease, specify the exact age of the donors. As the authors should know, aged MSCs have, in general, a limited differentiation capacity. Authors should take into account this observation, and mention in the manuscript that the obtained results are from aged MSCs. Was the differentiation capacity between the different donors similar? Moreover, could be RAR signaling pathway be affected in old MSCs and not in younger ones? Please discuss it. It could be an important point to be considered for a future therapeutic approach.
2) Gene expression data. The gene expression at 28 days of differentiation was compared versus day 0 or versus day 1? Page 10, lane 334, authors state that aggregated were harvested on day 1 (I suppose that this was used for the normalization), but in Figure legend they say that gene expression is normalized versus day 0. Moreover, authors state in page 10, lane 334, that aggregates were harvested at day 14 and day 28, but they only show gene expression data from day 28.
Importantly, there are no statistically significant results in the gene expression analyzed. Please, show these results with caution.
à It could be possible that the high standard deviation could be due to a high variability of gene expression between the different donors (high MSCs heterogeneity).
à The authors mention different housekeeping genes to perform the normalization. Why did they choose that genes (VPS29, PSMB4, REEP5)? Which was the normalization for each gene? Please, explain better this point. Have the authors considered use most common genes for normalization, such as GAPDH, or β-actin?
Page 10, lane 358: “Semiquantitative real-time…” If SYBRgreen was used, I supposed that the PCR was quantitative, that is, real-time PCR.
3) ALP activity. Please, specify how many aggregates were used to obtain medium supernatant. How did the authors normalize the ALP activity values? They should take into account that ALP activity depends on cell number. Was the cell number between different donors similar?
Reviewer 2 Report
In-vitro differentiated MSCs can undergo hypertrophy which limits the use of MSCs for cartilage tissue engineering applications. In this study, the authors aimed to attenuate the hypertrophy for possible application of these cells in tissue regeneration. Overall, this study is interesting but lacks key answers and supportive data.
General comments:
There are grammatical and spell mistakes throughout the manuscript. For example, in section 4.3, the correct word is “attenuation” and not “tennuation”. Similarly, figure legends are also not properly explained. In figure legends, each treatment group (classification group) should be well explained and not to use abbreviations as it makes it difficult to understand and focus on results.
Specific comments:
Figure 3: BMS treatment clearly shows the attenuation of hypertrophy. To demonstrate that cell size is affected by attenuating the hypertrophy, high magnification (20-40x) images should be taken and quantified for cell size. After the treatment of hypertrophic cells with BMS, do these cells proliferate ? it’ll be interesting to follow the proliferation in these treated cells and count the number. Figure 5: No significant differences in gene expression, hence difficult to make any conclusion. Gene expression should be presented as “fold change” Remove vertical lines between groups. In the discussion, the authors write “ We found statistically significantly higher ALP activity in the hypertrophic control group compared to the chondrogenic control group. But BMS treatment in both, early and late phase of differentiation, reduces ALP activity under hypertrophy enhancing conditions down to the range of the chondrogenic groups.” This statement is also not true and confusing. In figure 4A., BMS treatment both in the early and hypertrophic phase of chondrocytes slightly decreased ALP levels when compared to hypertrophic group only but not the level of chondrogenic group (Chon). Conclusion section: The authors state that “ when BMS was administered under hypertrophic conditions, a decrease in cell size, ALP activity and gene expression of hypertrophic markers is observed”, this statement is not supported by results, cell size was never measured and quantified, ALP and gene expression were also not significantly changed. Only supportive data reveals data in Figure 3.
Round 2
Reviewer 1 Report
The authors have properly addressed the suggested points.
Author Response
Thank you very much for your helpful advices.